# Recent Advances in Heterocyclic HIV Protease Inhibitors

**DOI:** 10.3390/ijms26189023

**Published:** 2025-09-16

**Authors:** Maria Funicello, Lucia Chiummiento, Alessandro Santarsiere, Francesco Poggio, Paolo Lupattelli

**Affiliations:** 1Department of Basic and Applied Science, University of Basilicata, via dell’ateneo lucano 10, 85100 Potenza, Italylucia.chiummiento@unibas.it (L.C.); alessandro.santarsiere@unibas.it (A.S.); 2Department of Chemistry, Sapienza University of Rome, p.le Aldo Moro 5, 00185 Roma, Italy; francesco.poggio@uniroma1.it

**Keywords:** AIDS, HIV-1 protease inhibitors, heterocycles, synthesis, biological activity

## Abstract

Since the first cases of AIDS, reported in 1980, this disease has become chronic over the years, and researchers have been trying to keep it under control. Despite the development and spread of mutate viruses, HIV protease remains an important pharmacological target. In the development of new HIV protease inhibitors, heterocyclic fragments have proven to be of great importance, owing to their rigid core structure, which may fit better into the enzyme’s hydrophobic pockets, and the presence of a heteroatom, which may increase the number of H-bonding interactions at the active site. According to the concept of targeting the protein backbone, different aromatic or non-aromatic heterocyclic moieties have yielded inhibitors with sufficient activity against mutant viruses. This paper provides an overview of HIV protease inhibitors developed over the last fifteen years, with a focus on the presence of heterocycles in their structure, either in the core or on the side chains, which are crucial for their activity. The rationale behind the design of these new inhibitors, as well as the key synthetic steps involved in their preparation, is also described.

## 1. Introduction

HIV-1 is the virus responsible for acquired immunodeficiency syndrome (AIDS), which is characterized by an attack and disruption of CD4+T cells, with the consequence of significant vulnerability in humans against different types of infection. This lentivirus integrates irreversibly into the host genome and has a long incubation period, being able to infect non-dividing cells [1,2].

Among the many strategies used to combat the disease, antiretroviral therapy (ART) is considered the most effective current treatment, and it is characterized by a combination of HIV protease inhibitors, reverse transcriptase inhibitors, and/or an integrase inhibitor. Recently, capsid inhibitors, such as Lenacapavir, have shown potential for a long-acting administration regimen [3].

HIV protease plays a crucial role in the viral lifecycle, being essential for the generation of mature, infectious viral particles [4]. This aspartyl protease is involved in the cleavage of peptide bonds in the gag and gag-pol polyproteins, and its inhibitors have been designed to bind to the active site with high affinity. All nine Food and Drug Administration (FDA)-approved inhibitors, except for tipranavir, are competitive peptidomimetic inhibitors, mimicking the natural substrate of the viral protease. Through the incorporation of the hydroxy ethylene core, the cleavage of the inhibitor is prevented. Due to their structural similarities, cross-resistance amongst inhibitors may occur, along with common side effects [5].

Although no new protease inhibitors have been approved or used in a clinical trial since 2007 [6], many small molecules have been described in the last fifteen years, with the aim of targeting the protein backbone. The goal has been to maximize interactions with enzyme backbone atoms [7,8]. Both extensive hydrogen bonding and hydrophobic interactions with enzyme subsites can limit drug resistance, as the geometry of the catalytic site must be conserved to maintain functionality [9,10,11]. In particular, the introduction of heterocyclic fragments has improved the activity of newly designed molecules, resulting in many potent inhibitors, as illustrated by FDA-approved darunavir and related compounds [12]. The introduction of cyclic scaffolds in the design of new chemical entities reduces flexibility and affords more rigid inhibitors. In particular, a cyclic scaffold can be envisaged as having either key interactions with catalytic aspartic acids or with residues belonging to the flap region of the active site [13]. The deep interest in cyclic chemotypes has been motivated by the different types of interactions formed with the enzyme while maintaining a key structural resemblance to a peptide substrate, to improve this profile for multidrug-resistant strains.

This review focuses on the main advances in the synthesis of new heterocyclic HIV protease inhibitors during the last 15 years and in the synthesis of key heterocyclic fragments [14]. The rationale behind the design of such new compounds is described, pointing out the crucial importance of the heterocyclic moiety for their activity. The different approaches are classified according to the type of heterocycle (aromatic or non-aromatic) and the heteroatoms involved (N, O, S, or polyheteroatomic). In the case of different heterocyclic moieties being present, the classification is made according to the group that is most critical for activity. This review aims to highlight ongoing research in the field of heterocyclic synthesis applied to the discovery of new HIV inhibitors.

## 2. Non-Aromatic Heterocycles

### 2.1. Non-Aromatic O-Heterocycles

Fused bicyclic *bis*-tetrahydrofuran (*bis*-THF) is critical for darunavir’s durable drug resistance profile. Due to its challenging double-ring structure with three chiral centers, many different synthetic approaches have been developed according to the source of the chiral carbons in *bis*-THF alcohol **2** (Figure 1) [15]. In this respect, enzyme-catalyzed resolution has been exploited by Khmelnitsky [16], who described the simple and efficient kinetic resolution of the racemic alcohol **2** using immobilized lipase to afford the desired optically pure (*R*)-*bis*-tetrahydrofuran (*bis*-THF) alcohol **3**. The reaction solvent, acyl donor, and immobilized biocatalyst proved to be critical factors.

The organocatalytic condensation of 1,2-dihydrofuran **5** (Figure 2) with glycolaldehyde catalyzed by Schreiner’s thiourea, coupled with enzymatic (Lipase PS) resolution, was described as a one-pot procedure by Itoh for the preparation of enantiopure **3** and applied to the environmentally friendly synthesis of darunavir [17].

A one-pot procedure using 1,2-dihydrofuran **5** and Cbz-protected glycol aldehyde as the starting materials was developed by Opatz through [2+2]-photocycloaddition between both reactants, followed by hydrogenation and lipase-catalyzed kinetic resolution, affording the target compound with a high yield and up to 99% *ee* (Figure 3) [18].

In 2020, Ghosh used the kinetic resolution of *bis*-THF alcohol by lipase (PS-30), in a late stage of the study as an alternative stereoselective approach to *cis* and *trans* 2,3-disubstituted tetrahydrofuran derivatives [19]. More recently, Hyster proposed a different chemoenzymatic approach to the use of *bis*-THF alcohol [20]. β-Ketolactone **8** was easily prepared and subjected to enantio- and diastereoselective dynamic kinetic resolution using suitable ketoreductase (KRED) and glutamate dehydrogenase (GDH) in phosphate buffer (KPi) from metagenomic mining (Figure 4). Subsequent lactone reduction with diisobutylaluminum hydride and phase-transfer cyclization afforded the desired fragment **3** in an acceptable yield.

Asymmetric catalysis was also exploited in the last period. Sudalai [21] proposed an efficient synthesis of (*S*)-tetrahydrofuran-3-ol **13**, which is a critical fragment in amprenavir and fosamprenavir.

Olefinic tosylate **10** was dihydroxylated under Sharpless asymmetric dihydroxylation conditions, furnishing, in a single step, the desired alcohol in good yield and moderate enantiomeric purity (Figure 5). Alternatively, epoxidation with mCPBA yielded the racemic epoxide **12**, which was subjected to co-catalyzed hydrolytic kinetic resolution, affording the alcohol along with the chiral epoxide in a lower chemical yield but with higher enantiomeric excess.

More and Ramana [22] described a Ru-catalyzed enantio- and diastereoselective dynamic kinetic resolution of benzyloxy/benzoyloxy-α-acyl-γ-butyrolactones *via* transfer hydrogenation, achieved through the in situ prepared (*R*,*R*)-Ru-FsDPEN catalyst. Starting from benzyloxy butyrolactone **14**, (3R,3aS,6aR)-hexahydrofuro[2,3–b]furan-3-ol **3** was prepared in good yield, with high *de* and *ee* through elaboration of the key intermediate (Figure 6).

Pd-catalyzed asymmetric hydroalkoxylation of ene-alkoxyallene, followed by ring-closing metathesis (RCM), was successfully employed by Rhee [23] for the preparation of hexahydrofurofuran-3-ol of type **E**, and was generalized to provide access to pyranofuranol and furopyranol derivatives in good yields and *ee* (Figure 7). For the synthesis of **3**, readily available allene **17** was coupled with commercial 2-bromo allylic alcohol **18** in the presence of Pd_2_(dba)_3_ as the palladium source and suitable ligand, affording the chiral adduct **19** in high yield and enantiomeric excess. Subsequent RCM provided the cyclic acetal **20**, which underwent a radical, mediated 5-*exo* cyclization to intermediate **21**, promoted by Bu_3_SnH in the presence of Et_3_B. Ozonolysis of this compound, followed by final reduction with NaBH_4_, furnished the target alcohol **3** in high yield with complete diastereoselectivity and no erosion of enantiopurity.

Organocatalytic methods were also successfully used. Ikemoto described an efficient synthesis of *bis*-THF alcohol **3** and its carbonyloxy-pyrrolidine-2,5-dione derivative taking advantage of diphenylprolinol-catalyzed enantio- and diastereoselective cross aldol reaction of polymeric ethyl glyoxylate **23** with the aldehyde **22** as the key step (Figure 8) [24]. Optimization of the reaction involved stirring polymeric ethyl glyoxylate in toluene solution with water prior to aldol reaction for an appropriate period. This pre-treatment was highly effective in accelerating the reaction when using 3 mol % of catalyst **A**. It also ensured excellent reproducibility, even when polymeric ethyl glyoxylate from different manufacturers was utilized. The aldehyde of intermediate **24** was then protected as dimethyl acetal and the ester was reduced, yielding diol **26** in an overall yield of 85%. An acetal exchange reaction catalyzed by H_2_SO_4_ followed by hydrogenation with Pd/C catalyst afforded **3** in quantitative yield over two steps. Shortly thereafter, the same group described a series of amino perfluoroalkanesulfonamide derivatives of diarylprolinols, which proved to be effective organocatalysts [25].

Traditional chiral-substrate approaches have also been employed over the past fifteen years. A spirocyclic dioxolane derivative from *D*-glyceraldehyde **27** was used by Kulkarni as starting chiral substrate for the preparation of all four isomers of bis-tetrahydrofuran alcohol in good overall yield (38%) of the active isomer **3** (Figure 9) [26].

(S)-Glyceraldehyde was used as a chiral source by Ghosh in the convergent synthesis of various substituted bis-THF derivatives, to find new HIV-protease inhibitors with enhanced binding capacity [27]. The synthesis of the *bis*-THF ligand began with known compound **30** (Figure 10) prepared in multigram quantities by Wittig olefination of (S)-glyceraldehyde acetonide with (ethoxycarbonylmethylene) triphenylphosphorane. DIBAL-H reduction yielded the corresponding alcohol **31** in nearly quantitative yield. Subsequently, *O*-alkylation and subsequent stereoselective [3,4]-sigmatropic rearrangement afforded the alcohol with three chiral centers **33**, which was transformed alternatively to *bis*-THF alcohol **3** or to different functionalized derivatives. The synthesis of alkyl substituted bis-THF ligands **36** required a Mitsunobu reaction on alcohol **33** with PPh_3_/DIAD and *p*-nitrobenzoic acid, to give the corresponding ester **34**. This was first hydrolyzed, then alkylation with BnBr or MeI was performed, alternatively. Final reduction, ozonolysis, and cyclization sequence furnished the ligands **36** in good overall yield. Among the inhibitors bearing new *bis*-THF fragments, compound **37** resulted in being the most potent.

More recently, Ghosh described a new approach to bis-THF alcohol **3**, starting from commercially available sugar derivatives [28]. As an example, commercial 1,2-*O*-isopropylidene-α-D-xylofuranose **38** was converted to α,β-unsaturated ester **39**, by sequential selective protection of primary alcohol, Swern oxidation, and Wittig olefination (Figure 11). Ester **39** was then submitted to highly stereoselective substrate-controlled catalytic hydrogenation, which represented one of the key steps.

Saturated ester **40** was converted to γ-lactone derivative **41** by exposure to BF_3_·OEt_2_ followed by Et_3_SiH. Hydrolysis of the benzoate furnished the corresponding bicyclic alcohol, which was converted to methyl acetal **42** by a three-step sequence involving Dess–Martin oxidation of the primary alcohol to the corresponding aldehyde, Baeyer–Villiger oxidation promoted by *m*-CPBA, and exposure of the resulting formate to 6% HCl in MeOH. Acetal **42** was obtained in 35% yield over four steps. Final reduction of the lactone and acidification yielded the target bis-THF alcohol.

Regarding sugars as chiral pool materials, Sridhar developed a stereoselective synthesis of carbohydrate-derived perhydrofuro[2,3-*b*]furan derivatives, starting from sugar-derived allyl vinyl ethers [29]. In particular, *bis*-THF alcohol **3** was prepared using 3,4-di-*O*-acetyl-D-arabinal **43** as the chiral starting material (Figure 12). It was first subjected to Ferrier rearrangement with 2-(phenylselenyl)ethanol, which provided 2,3-unsaturated glycoside **44**. After oxidation to selenone **45**, a base-mediated thermal fragmentation furnished the allyl vinyl ether **46**, which underwent a Claisen [4]-sigmatropic rearrangement, affording the expected 3-C branched derivative **47**. Reduction of the aldehyde, followed by ozonolysis of the olefin and acid-mediated acetalization, provided perhydrofuro[2,3-*b*]furan **50** in good yield. The synthesis of *bis*-THF alcohol **3** required deprotection of the ester, Mitsunobu inversion with *p*-nitrobenzoic acid in PPh_3_/DIAD system, and final hydrolysis.

The potassium isocitrate, obtained from high-yielding fermentation fed by sunflower oil, was the starting material for the preparation of bis-THF alcohol **3** in the work of Yue (Figure 13) [30]. After several steps, it was transformed into a tertiary amide, which was reduced along with the ester functionalities to a transient aminal-triol. This latter was converted in situ to the desired *bis*-THF alcohol. Each step was optimized, resulting in the key alcohol (in the form of activated carbonate) with an overall yield of 43%. Considering the low cost of the starting material and the efficiency of the overall synthesis, the method appeared to be effective in reducing the cost of darunavir.

With the aim of creating a new class of inhibitors with improved pharmacological and drug-resistance profiles, Ghosh described the synthesis of a 6–5–5 ring-fused crown-like tetrahydropyranofurans and their incorporation into new potent inhibitors [31,32]. The synthesis started with an asymmetric Diels–Alder reaction between **57** and cyclopentadiene, using a chiral oxazaborolidinium cation **A**, as a key step (Figure 14). The cycloadduct was oxidized with H_2_O_2_ to afford alcohol **59** in 80% yield over two steps, with 98% *ee*. After protection as the TBS ether, the intermediate was converted to bicyclic acetal **60** in three-steps, involving reduction with LiAlH_4_, one-pot oxidative cleavage of the olefin, and reduction of the resulting aldehyde with DIBAL-H. The lactol **60** (a 2:1 mixture) was treated with trifluoroacetic acid (TFA) to produce the bridged tricyclic derivative **61**, and the TBS group was removed with tetrabutylammonium fluoride (TBAF), yielding the desired *exo*-alcohol. This compound was alternatively epimerized by Dess–Martin oxidation followed by reduction of the resulting ketone with NaBH_4_. The inhibitor **64** displayed superior antiviral activity and drug-resistance profiles compared to darunavir.

Shortly afterwards, this fragment was incorporated into new arylboronic derivatives, and they were tested against highly drug-resistant HIV-1 variants [33].

Alternatively, the Diels–Alder reaction of cyclopentadiene with chiral 3-(acyloxy)acryloyloxazolidinone derivatives was later used for the synthesis of 6-5-5 fused crown-like THF ligands (Figure 15). In particular, 3-(4-methoxybenzoyl)acryloyl oxazolidinone derivative **65** and cyclopentadiene afforded the endo-diastereoselective derivative **66** in 98% yield. Reaction of the cycloadduct in MeOH provided the corresponding methyl ester **67** in excellent yield, which was a key intermediate in the synthesis of the bicyclic alcohol **63** [34].

An optically active (1*R*, 3aS, 5*R*, 6*S*, 7*aR*)-octahydro-1,6-epoxy-isobenzo-furan-5-ol derivative was recently described as a high affinity ligand for HIV-1 protease inhibitors [35]. The stereoselective synthesis was based on enantioselective enzymatic desymmetrization of *meso*-1,2(dihydroxy methyl)cyclohex-4-ene **69** using porcine pancreatic lipase (PPL) to produce optically active alcohol **70** (Figure 16) [28]. It was then converted to a functionalized cyclohexene derivative **71**, which underwent Sharpless asymmetric dihydroxylation to yield a 1:1 mixture of diastereomeric diols **72** and **73**. The diols were separated and only diastereomer **73** was utilized for the synthesis of ligand alcohol **74**. In general, inhibitors bearing such ligands showed very potent activity. Moreover, compound **75**, with a difluorophenylmethyl P1 ligand and an aminobenzothiazole as the P2′ ligand, maintained high potency against a panel of highly multidrug-resistant HIV-1 variants.

More recently, a modified procedure was described [36]. Optically active alcohol **70** was converted to methyl acetal **76** (in 75% yield over 3 steps) as the major isomer in a sequence involving, (1) Swern oxidation of alcohol **70** to the aldehyde, (2) treatment of the resulting aldehyde with CH(OMe)_3_ in the presence of tetrabutylammonium tribromide (TBABr_3_), followed by treatment with 1 N NaOH, and (3) reaction of the resulting methyl acetal with a catalytic amount of CSA (Figure 17).

Stereoselective *syn*-1,2-diol functionality was introduced on methyl acetal **76** by substrate-controlled dihydroxylation, with K_2_OsO_4_/K_3_Fe(CN)_6_ system, affording *syn* diol **77**, as 95/5 diastereomeric mixture. Swern oxidation provided 1,2-diketone derivative **78**, which was reduced with L-Selectride to give the corresponding inverted *syn*-1,2-diol, stereoselectively. Acid-catalyzed cyclization then furnished tricyclic ligand alcohol **74** with high enantiomeric purity.

The synthesis of enantiomerically pure (3aS,4S,7aR)-hexahydro-2H-furo[2,3-b]pyran-4-ol **83** was originally described by Ghosh [37] starting from known enantiomerically pure lactone **80** (Figure 18). Subsequent steps involved reduction, selective protection/deprotection, ozonolysis, and reduction of the hemiacetal moiety. Final removal of the silyl group with TBAF furnished the desired ligand.

A more recent alternative route has been proposed, starting from inexpensive materials and taking advantage of highly enantioselective enzymatic desymmetrization of a *meso*-diacetate, already developed by the same group [38]. The enzymatic desymmetrization of **84** was optimized by using aqueous 1 N NaHCO_3_ for neutralization of acetic acid formed during the reaction (Figure 19). Swern oxidation followed by protection of the aldehyde furnished the corresponding acetal ***ent*-71**. Oxidative cleavage of the olefin by ozonolysis in the presence of pyridine as the organocatalyst, followed by reduction of the resulting dialdehyde with sodium borohydride afforded diol **85**. Treatment of the diol with a catalytic amount of CSA furnished bicyclic acetal **86** in 56% yield over 3 steps. After hydrolysis of acetate, the corresponding alcohol was treated with *o*-nitrophenylselenonitrile and *n*-tributylphosphine to give the corresponding selenide derivative. Oxidation of the resulting selenide with *m*-CPBA resulted in elimination of the corresponding selenoxide to afford olefin **87** in 71% yield over 3 steps. Ozonolysis of *exo*-olefin followed by reduction of the corresponding ketone yielded alcohol **83** in good yield.

Various C3-functionalized cyclopentanyltetrahydrofurans (Cp-THF) were prepared and combined with hydroxyethylsulfonamide isosteres to obtain new inhibitors with high antiviral activity, including against a panel of multidrug-resistant HIV-1 variants [39]. As an example, 3-(*R*)-acetoxy and 3-(*R*)-methoxy ligands **92** and **93** were prepared from the known alcohol **89**, which was first alkylated to provide methyl ester **90** (Figure 20). DIBAL-H reduction followed by radical cyclization provided 3-(*R*)-hydroxy derivative **91** in 10:1 diastereomeric ratio. The major isomer was transformed into the desired ligand **92** by acetylation, followed by removal of the silyl group, in 73% yield over two steps. Alternatively, the methoxy derivative **93** was prepared by methylation followed by removal of silyl group, in 71% yield.

Novel oxatricyclic ligands were designed and synthesized by Ghosh to enhance interactions with the protease backbone [9]. In particular, the inhibitor **101**, bearing the tris-THF *syn*-*anti*-*syn* configuration, showed very high activity against a variety of multidrug resistant HIV-1 variants and blocked HIV-protease dimerization about 10-fold better than darunavir (Figure 21) [40]. Starting from the known acetate **94**, the bicyclic enol-ether **95** was easily obtained. Subsequently, after epoxidation with dimethyldioxirane (DMDO) and regio- and stereoselective oxiranyl ring opening with sodium methoxide, the desired *endo*-alcohol **96** was obtained by Dess–Martin oxidation followed by L-selectride reduction of the resulting ketone. Acylation of **96**, followed by treatment with propargyl alcohol, afforded acetals **97** and **98** in a 1:4 ratio. After removal of the acetate, the major diastereomeric alcohol deriving from **97** was converted into the corresponding tricyclic olefin **99**. Ozonolysis followed by reduction furnished *syn*-*anti*-*syn* type oxatricyclic tris-THF **100** as a single isomer.

*Gem*-difluoro-bis-THF ligands were prepared and converted into suitable inhibitors, which exhibited very high activity and better lipophilicity profiles than darunavir, along with significantly improved blood–brain barrier permeability in an in vitro model [41]. The synthesis began with DIBAL-H reduction of optically active methyl ester **102** (Figure 22), producing dibenzyl-L-glyceraldehyde. This intermediate underwent the Horner–Emmons reaction with sodium hydride and triethyl phosphonoacetate, affording the corresponding α,β-unsaturated ester in 88% yield over two steps. Reduction with DIBAL-H provided the corresponding allylic alcohol, which was treated with chlorodifluoroacetic acid in chloroform at reflux, yielding difluoroacetate derivative **103** in 90% yield over two steps. Subsequently, it underwent a Reformatskii–Claisen reaction by treatment with trimethylsilyl chloride and activated zinc dust. The subsequent acid-catalyzed esterification furnished a 2:1 mixture of diastereomers in 80% yield over two steps. The mixture could be separated after its conversion into the Weinreb amides, by treatment with HN(Me)OMe·HCl and *n*-BuLi. These amide diastereomers were separated by silica gel chromatography, providing the syn diastereomer **104** as the major product and the anti diastereomer **105** as the minor product, in a 2:1 ratio and with an overall yield of 80%.

Reduction of the Weinreb amide **105** was carried out with lithium aluminum hydride, and the resulting crude aldehyde was reduced with sodium borohydride in a one-pot operation to provide difluoro alcohol **106** in near quantitative yield. Ozonolysis, followed by reductive cleavage with PPh_3_, provided cyclic acetal **107** upon cyclization. Final catalytic hydrogenation and treatment with CSA of the triol afforded difluoro-*bis*-THF **108**.

To maximize the ligand-binding site interactions in the protease active site, Ghosh described cyclohexyl-derived ligands within a 6-5-5 fused ring system [42].

Starting from 1,3-diketone **109**, reaction with dihydrofuran **110** in the presence of Mn(OAc)_3_·2H_2_O afforded the corresponding tricyclic derivative **111** (Figure 23). Enone **111** was first hydrogenated, and the resulting ketone was reduced with NaBH_4_, to give racemic *endo* alcohol **112**. This racemic alcohol was subjected to enzymatic resolution using lipase PS-30, which provided the optically active acetate derivative **114** (45%yield) and alcohol **113** (45% yield). Acetate **114** was converted back to the alcohol **113** in 89% yield by transesterificaton using NaOMe in MeOH. A series of ligands was prepared and incorporated into new inhibitors. Notably, compound **115** exhibited very impressive enzyme and antiviral potency (Ki = 10 pM, antiviral IC_50_ = 1.9 nM).

Novel isosorbide scaffolds were proposed by Liu as P2 ligands of HIV-1 protease inhibitors bearing the hydroxyethylamine core **116** (Figure 24) [43]. They showed very high inhibition activity with IC50 in the nanomolar or picomolar range. From a preparative perspective, they leveraged the commercial availability of starting isosorbide mononitrate **117**, which was easily elaborated and activated as a mixed carbonate. Specifically, it was transformed into mixed carbonate **118** with *p*-NO_2_PhOCOCl in good yield or converted into TBS-protected alcohol **119** via TBS ether formation, followed by cleavage of nitrate under reductive conditions. The alcohol was then methylated; the TBS group was cleaved, and final carbonate formation afforded the key intermediate **120**.

Chiral 4,4-dimethyltetrahydrofuran-3-ol was prepared by Srinivasa Reddy and used for the synthesis of new amprenavir analogs [44]. Starting from pantolactone **122** (Figure 25), which is available in both enantiomeric forms, lactol **123** was obtained by protection and reduction. Subsequent reduction with BF_3_Et_2_O and triethylsilane, followed by final deprotection with TBAF, afforded the desired THF-alcohol.

Different *O*-heterocycles were also proposed by Yajima as HIV-1 protease inhibitors [45]. Fornicin A, a meroterpenoid with a γ-butylolactone moiety in its side chains showed weak anti-HIV-1 protease activity without cytotoxicity against E-PR293 cells. The synthesis involved the preparation of optically active alcohol **130** from the starting material 2,5-dihydroxybenzaldehyde **126** (Figure 26), using Sharpless asymmetric dihydroxylation on the suitable benzothiazolsulfide **129** as the key step. Oxidation with *m*CPBA and a final Smiles rearrangement of β-hydroxysulfone yielded the desired alcohol **130**, which was coupled with the suitable acid segment prepared from the known aldehyde **131**. Ring-closing metathesis using Grubbs 1st catalyst and a final deprotection step furnished Fornicin A.

Flexible macrocycles between the P1′-side chain and a suitable P2′-ligand were proposed by Ghosh to broaden the activity of the inhibitor by achieving a better fit in the S2 hydrophobic pocket, which increases in size upon certain mutations [46]. New inhibitors bearing 16- to 19-membered macrocyclic rings, connecting a nelfinavir-like P2 ligand and a tyrosine side chain containing a hydroxyethylamine sulfonamide isostere, were synthesized and found to be more potent than their corresponding acyclic counterparts. In particular, compound **134** showed the best enzyme inhibitory and antiviral activity (K_i_ = 0.2 nM, IC_50_ = 0.21 mM) (Figure 27). The synthesis of the desired tyrosine-derived hydroxyethylamine sulfonamide isostere **138** began with butadiene monoxide **135**, which was transformed into the corresponding allylic alcohol. This was subjected to Sharpless asymmetric epoxidation, and the resulting epoxide **136** was regioselectively opened by TMSN_3_/Ti(O-*i*Pr)_4_ system to afford the corresponding azido diol, which was subsequently converted to epoxide **137**. After introducing the sulfonamide moiety, a Boc-protected amine and a free phenol were formed in *one pot* process via catalytic hydrogenation in the presence of Boc_2_O. Finally, the introduction of an allyl chain furnished the desired intermediate **138**.

The P2 fragment was synthesized as 3-hydroxy-2-alkenylbenzoic acid from maleic anhydride **140** via formation of phosphorane **141**, followed by Wittig reaction with known aldehyde **142**. The resulting dienoic acid was then alkylated and subjected to 1,6-electrocyclization by exposure to TFAA/Et_3_N, followed by reduction with NaBH_4_ (Figure 28). Final saponification yielded the desired benzoic acid **144**. After amine deprotection and coupling with the benzoic acid, the resulting acyclic diene was subjected to ring-closing metathesis (RCM) with Grubbs’ 1st generation catalyst, furnishing the desired macrocycles.

### 2.2. Non-Aromatic N-Heterocycles

Among non-aromatic *N*-heterocycles, the structure of 1,3-diazacycloalkan-2-ones is of interest due to its presence as the core of HIV-protease inhibitor DMP 450 (Figure 1).

Frain described the synthesis of novel tetrahydropyrimidinones, starting from tetraols, that are easily prepared from suitable alditols [47]. Tetraol **146**, from D-(+)-arabitol, was first protected in primary hydroxyl groups as TBS ethers (Figure 29). The introduction of the amines was performed by mesylate activation of alcohols, followed by displacement with azide and reduction by hydrogenation. Extended hydrogenation periods led to the isolation of the mono TBS-protected diaminodialcohol **148**. Subsequent reaction with carbonyl diimidazole in the presence of base provided the crude tetrahydropyrimidinone, which was treated with acid to remove the TBS group. Final perbenzylation furnished the functionalized tetrahydropyrimidinone **150**, which exhibited modest HIV-protease inhibition activity.

Unsymmetrical 1,3-diazacycloalkan-2-ones were prepared by Jamir [48] from thiocarbamate salts using sodium percarbonate as an oxidant. These compounds were evaluated as HIV-1 protease inhibitors via in silico approach. Secondary amines **152**, such as morpholine, piperidine, 4-hydroxy piperidine, diethylamine, and di-isopropylamine, were reacted with phenyl isothiocyanate and other substituted phenyl isothiocyanates **151** to afford the corresponding unsymmetrical 1,3-diazacycloalkan-2-one in high yields (Figure 30). Computational assessment of IC_50_ values using known references satisfactorily confirmed the inhibitory activity of the selected compounds against HIV-1 protease.

In the work of Jadhav [49], *bis*-allylidene-4-piperidones analogues were prepared by Claisen–Schmidt condensation between 2,6-substituted piperidin-4-ones and cinnamaldehyde in a basic medium (Figure 31). In particular, 3,5-*bis* (3-phenyl-allylidene)-2,6-diphenyl-piperidine-4-one **155** was obtained from the parent compound **154**. The *N*-tosyl derivative **156** was prepared by reacting **155** with tosyl chloride. Compound **156** showed moderate HIV-1 protease inhibitor activity compared to the standard, pepstatin-A.

Indolin-2-one moieties were proposed by Eissenstat to interact with the S20 subsite of the HIV protease binding pocket [50]. Several of these inhibitors were synthesized and exhibited sub-nanomolar Ki values and antiviral IC50s in the low nM range against wild-type (WT) HIV and a panel of multi-drug resistant (MDR) strains. The darunavir sulfonamide was replaced by an indolin-2-one moiety, reacting *bis*-THF amino analog **157** with indolin-2-one sulfonyl chloride **158**, affording compound **159**. Subsequent heating in the presence of DMF dimethylacetal or related acetals led to the formation of indolin-2-one **160** (Figure 32). The amine portion of the enamines could be readily varied through an exchange reaction of dimethamino analog **161** with excess amine.

1,4-benzodiazepine derivatives have been recognized as mimicking the β-hairpin flap of HIV-1 protease, thereby interfering with the flap–flap protein–protein interaction which controls access to the active site. Recently, the Konvalinka group screened a variety of chemical structures for inhibition of HIV replication [51]. This led to the identification of several new compounds that inhibited HIV PR in the low micromolar range. In particular, compound **166** resulted in being the most potent and selective for HIV PR. The monomer was prepared by reacting *o*-phenylenediamine **162** with dimedone in the presence of catalytic amount of TFA, affording the corresponding enamine **164** which was then reacted with benzaldehyde to obtain monomer **165**. Treatment with oxalyl chloride at low temperature, furnished the desired dimer **166** in modest yield as a diastereoisomeric mixture (Figure 33), whose racemate (11*R*, 11′*R* + 11*S*, 11′*S*) showed inhibition activity in the nanomolar range (IC_50_ = 30 nM).

New bioactive compounds containing 1,4-benzodiazepine scaffolds were proposed by Fletcher, with the most potent compound, **167**, inhibiting the protease with a modest Ki of 11 µM (Figure 34) [52]. It was prepared starting with regioselective nitration of benzoic acid **168**, followed by benzylic oxidation, which afforded ketone **169**. After esterification, catalytic hydrogenation yielded aniline **170**. Amide **171** was prepared coupling aniline **170** with Fmoc-protected pre-activated D-Leu, and then subjected to base-mediated cyclization, furnishing the key benzodiazepine nucleus. After deprotection and *N*-alkylation, the final compound **167** was obtained. Different compounds were obtained by deprotection and/or alkylation. Methyl imine could be replaced by a phenyl imine, resulting in the formation of different compounds.

The tetrahydropyridine ring system is a widely distributed structural framework involved in numerous pharmaceuticals and natural products. Mohammadi [53] recently described an efficient, *one-pot*, catalyst-free, four-components procedure for the synthesis of novel 10β-hydroxy-4-nitro-5-phenyl-2,3,5,5a-tetrahydro-1*H*-imidazo[1,2-a]indeno[2,1-e]pyridin-6(10b*H*)-one derivatives **177** from the corresponding diamine **173**, nitro ketene dithioacetal **174**, aryl aldehydes **175,** and 1,3-indandione **176** (Figure 35). The overall transformation consisted of a Knoevenagel condensation, Michael addition, tautomerism, and cyclisation sequence. All newly synthesized compounds were screened for molecular docking studies. Some of them showed minimum binding energy and good affinity toward the active pocket of HIV protease enzyme compared to Saquinavir.

A flexible piperidine moiety was introduced into a novel class of HIV-1 protease inhibitors as the P2 ligand by Wang Y., Cen, and Wang J [54]. In particular, inhibitor **181**, which features (*R*)-piperidine-3-carboxamide as the P2 ligand and 4-methoxybenzenesulfonamide as the P2′ ligand, showed more than a 6-fold enhancement of activity compared to darunavir (IC_50_ 0.13 ± 0.01 nM). Furthermore, there was no significant change in potency against darunavir-resistant mutations and HIV-1_NL4-3_ variant. For the synthesis, easily prepared (*R*)-piperidine-3-carboxylic acid **178** was reacted with (Boc)_2_O to obtain the corresponding Boc-amino derivative **179** in excellent yield. Coupling of the acid with a suitable amine, under the catalysis of EDCI, HOBt, and DMAP, furnished the amide **181** (Figure 36). Removal of the Boc group provided the target compound.

Macrocyclic peptides have emerged as a new class of drug discovery modalities because they are considered more likely to acquire strong interactions with target proteins, owing to their restricted molecular motion and rigidity compared to linear peptides. Recently, Mikamiyama [55] discovered a novel HIV-1 protease inhibitor, **182** (Figure 2), with potent antiviral activity (EC_50_ 37 nM) and oral bioavailability, using a structure-based drug design approach via X-ray crystal structure analysis. This approach started from hit macrocyclic peptides identified by mRNA display against HIV-1 protease. In particular, the improvement of the proteolytic stability of macrocyclic peptides by introducing a methyl group at the *α*-position of certain amino acids, such as proline, is crucial for exhibiting strong antiviral activity.

The five membered hydantoin scaffold was already recognized as a key fragment in HIV protease inhibitors. Recently, Zhang reported a new general iridium-catalyzed asymmetric hydrogenation of hydantoin and thiazolidinedione-derived exocyclic alkenes using BiphPHOX as a ligand [56]. The transformation, which showed good functional group tolerance, high yields, and enantioselectivities in the hydrogenated products, was applied to the synthesis of **189**, a key intermediate in the preparation of an HIV protease inhibitor **190** (Figure 37) [57]. A gram-scale reaction was carried out starting from alkene **186**, yielding the hydrogenated product **187** with 92% *ee* and 97% yield. This product was then reduced with LiAlH_4_, providing chiral 1,3-diazacycloalkan-2-one **188**. The reduced product **188** was transformed into the key intermediate **189** + dechlorination in the presence of Pd/C.

A densely functionalized iminohydantoin, bearing a quaternary stereocenter, was recently described and developed as an HIV protease inhibitor [58,59]. Ischay and Hoang described process development efforts that enabled the first scale-up of compound **196** (Figure 38). Several challenges were addressed, including reaction optimization, purification, stabilization of the intermediates, and deprotection procedures. For the reaction of iminohydantoin **195** formation, the addition order was evaluated. It was found that the presence of the amine coupling partner, in the form of BSA (benzensulphonic acid) salt, was required during the loading of the EDC·HCl reagent. Reagent stoichiometry was then investigated, showing that the loading of EDC·HCl could be reduced to 1.5 equivalents and 6·BSA to 1.1 equivalents without negatively affecting reaction performance. The use of 2–3.5 equiv. *i*-Pr_2_NEt proved optimal for scale-up to 687 g. A MIBK (methyl isobutylketone) solvate was identified as a crystalline form that provided an additional isolable intermediate, serving as a purity control point prior to Cbz-group removal and API isolation. The Cbz protecting group was removed from iminohydantoin **195** using HCl in acetic acid, that mitigated hydrolysis of the iminohydantoin. After work-up, crystallization afforded **196** 1-succinate (Figure 38).

Wang and coll. recently reported a new strategy for the synthesis of an iminohydantoin derived from chiral quaternary *α*-aryl amino acids [60]. Such fragments are present in certain HIV protease inhibitors [58]. They took advantage of newly developed chiral Karady–Beckwith dehydroalanines, which were prepared for use in a photoredox-mediated highly stereoselective Giese-type reaction with carboxylic acids and tertiary amines. A final stereoselective Clayden rearrangement afforded chiral quaternary *α*-aryl amino acid derivatives. A new chiral dehydroalanine bearing an *N*-methyl, *N*-phenyl-urea moiety **202** was synthesized starting from *S*-benzyl-L-cysteine **197**, which was first transformed into amino amide **198** (Figure 39). Cyclization with pivalaldehyde was then performed, and the free NH group was converted into the corresponding urea moiety, affording intermediate **201**. An exocyclic double bond was obtained by oxidation of sulfide and subsequent elimination of the sulfoxide. Compound **202** underwent a Giese-type reaction with pivalic acid under the irradiation with a blue light-emitting diode (LED), using 4CzIPN as the photocatalyst (PC) and Cs_2_CO_3_ as the base in DMF, affording the desired adduct **203** with a high diastereoisomeric ratio (>20:1). This intermediate smoothly underwent a Clayden rearrangement, giving rise to the quaternary amino acid derivative **204** in high yield and diastereoselectivity. Final deprotection afforded free amino acid **205**, suitable for incorporation into the HIV protease inhibitor **206**.

### 2.3. Non-Aromatic Heterocycles with Multiple Heteroatoms

Starting from KNI-764-derived isostere prepared by Mimoto and coworkers [61] Ghosh [62] proposed a new series of protease inhibitors in which cyclic ethers were incorporated as P2-ligands. As an example, **GRL-0355** (Figure 40) displayed impressive antiviral properties, with improved potency and efficacy against multidrug-resistant HIV-1 variants. The key intermediate **207**, prepared through two alternative pathways, was protected as the corresponding acetonide, after which the acetate group was hydrolyzed to afford the corresponding alcohol. This intermediate was subjected to oxidation using ruthenium chloride hydrate and sodium periodate, resulting in the formation of the target carboxylic acid **208**. Compound **208** was then transformed into amide **214** by activation to the corresponding mixed anhydride, followed by reaction with heterocyclic amine **213** [63]. The latter was derived from (*R*)-BocDMTA **212**, which was synthesized with 99.4% *ee* via enantioselective hydrolysis of methyl (±)-5,5-dimethyl-1,3-thiazolidine-4-carboxylate **210** by a *Klebsiella oxytoca* hydrolase.

The work of Guarna and his group on small peptidomimetics bearing 6,8-dioxa-3-azabicyclo[3.2.1]-octane scaffold resulted in the identification of HIV protease inhibitors with IC_50_ values in the sub-micromolar range [64]. They proposed the bicyclic acetal portion as a potential transition state analog in interactions with the enzyme active site. In particular, the best results were obtained with the glycine-derived scaffold **BTG(O)**-**A** (bicyclices derived from tartaric acid and glycine), which bears an α-amino alcohol fragment at position 7, and with phenylalanine-derived scaffold **BTF(O)-B** (bicyclices derived from tartaric acid and phenylalanine), which bears small aliphatic chains at the same position.

The synthesis began with benzylamine **215**, which was treated with bromoacetaldehyde dimethylacetal. Amine **216** was then acylated with (2*R*,3*R*)-di-O-acetyltartaric anhydride, yielding amide **217**. This was subsequently transformed into cyclic acetal **218** by treatment with MeOH/HCl. The 6,8-dioxa-3-azabicyclo[3.2.1]-octane **219** was obtained by treating **218** under reflux in toluene in the presence of H_2_SO_4_ over silica gel. Amide–ester exchange reactions allowed the preparation of the desired amides **BTG(O)-A** (Figure 41).

A novel HIV protease inhibitor was designed by Bungard using a morpholine core as the aspartate binding group [65]. Analysis of the crystal structure of the initial lead bound to HIV protease enabled optimization of enzyme potency and antiviral activity. This resulted in a series of potent orally bioavailable inhibitors, among which **MK-8718** (Figure 42) was identified as a compound with a favorable overall profile. The preparation of a suitable morpholine intermediate started with the reaction of amino alcohol **220**, derived from D-serine, with (*R*)-epichlorohydrin to give morpholine **221**. Swern oxidation afforded aldehyde **222.** Subsequently, Seyferth–Gilbert homologation yielded alkyne **223**. Sonagashira coupling, followed by alkyne reduction, gave aniline **224**. Cbz protection, followed by TBS deprotection, afforded the corresponding alcohol. Carbamate formation, followed by Cbz-deprotection, produced the desired aniline **225**. Final elaboration by coupling with a suitable diaryl-α-aminopropanoic acid furnished **MK-8718**.

Morpholine was also proposed by Zhou, Cen, and Wang [66] as a P2 ligand of HIV-1 protease inhibitors with a flexible heterocyclic structure, which can fit into the minimally distorted active site of darunavir-resistant HIV-1 variants. Starting from easily available *N*-substituted morpholines, a new series of inhibitors was synthesized and tested. As an example, 2-Morpholinoethan-1-ol **226** was activated as carbonate, then a suitable aminoalcohol fragment was alkoxycarbonylated, providing carbamate inhibitor **229**, which exhibited almost 4-fold superior activity against wild-type HIV protease, compared to darunavir, along with appreciable antiviral activity against darunavir-resistant HIV-1 variants (Figure 43).

### 2.4. Heteroaromatics

Based on Ritonavir, Kaye designed novel, structurally simplified, truncated analogs bearing heteroaryl groups linked to the peptidomimetic chain [67]. Starting from pyridine-2-carbaldehydes **230**, the indolizine-2-carboxylate esters **232** (Figure 44) were synthesized by Baylis–Hillman reaction followed by cyclization promoted by acetic anhydride at reflux, through acetylation of the alcohol, which facilitated the reaction. Saponification of the esters yielded the corresponding carboxylic acids **233**. Different amides were prepared by coupling with suitable amines in presence of carbonyl diimidazole (CDI). According to the authors, inhibition activity was not measured; however, enzyme-binding, enzyme inhibition, and in silico docking studies of those compounds were planned.

The incorporation of substituents with hydrogen bond donor and acceptor groups at the P1 position of a symmetry-based HIV protease inhibitor series was described by DeGoey and resulted in significant improvement in potency against the resistant mutants [68]. Overall, compound **240** (Figure 45) demonstrated the best balance of potency against drug resistant strains of HIV and exhibited oral bioavailability in pharmacokinetic studies. The synthesis began with benzylated tyrosine **234** and reaction with sodio acetonitrile yielded nitrile **235**, which was then treated with a benzyl Grignard reagent to give the enaminone **236**. A high degree of stereocontrol was observed during the stepwise reduction of **236** with NaBH_4_, followed by Boc protection of the resulting amine. Removal of the benzyl protecting groups through hydrogenolysis gave the amine **238**. The corresponding triflate was generated using *N*-phenyl-bis(trifluoromethanesulfonimide) affording **239**. Palladium-mediated coupling with 5-fluoro-3-pyridylboronic acid furnished the bis aryl intermediate, which was finally coupled with a suitable *t*-butyl-glycine derivative, yielded the desired inhibitor **240**.

During the study of novel inhibitors against South African wild-type (C-SA) HIV protease, Makatini [69] described the first pentacycloundecane (PCU) diol peptoid-derived inhibitors, with IC_50_ values ranging from 6.5 to 0.075 µM. Starting from inhibitor **241**, which showed very good activity (IC_50_ 0.075 µM), it was derivatized by substituting the carbobenzyloxy group with the (2-pyrimidinylthio)acetic acid group in order to produce compound **242** for improved solubility (Figure 46). Unfortunately, this molecule exhibited significantly less binding affinity to the enzyme (IC_50_ 1.0 µM).

The discovery of HIV-1 protease inhibitors, which exhibit high potency against both HIV-1 wild-type and multi-PI-resistant HIV-mutants is always a key focus. The work of Kesteleyn [70] was oriented towards discovering new PIs suitable for a long-acting injectable drug applications. In this regard, they described new compounds bearing a heterocyclic 6-methoxy-3-pyridinyl (MP) or a 6-(dimethylamino)-3-pyridinyl (DMAP) (R_3_) group at the *para*-position of the P1′ benzyl fragment, which showed antiviral activity in the low nanomolar range. The introduction of a heteroaryl moiety was performed via a Suzuki coupling reaction on polyfunctionalized bromoaryl intermediate **249**. Bromo lactone **248** was recognized as a key intermediate. For the synthesis of **248**, a new enantioselective methodology was developed. As shown in Figure 47, the first step involved converting *N*-Boc-protected (S)-4-bromophenylalanine **243** into the Weinreb amide **244**. Treatment with 3-butenylmagnesium bromide then furnished ketone **245**, which was oxidized at the terminal double bond to afford the γ-ketocarboxylic acid **246**. The desired bromo-lactone **247** was obtained by initially esterifying the carboxylic acid and then performing a reductive cyclization with *N*-selectride. Final benzylation, using lithium hexamethyldisilazane as a base at low temperature, produced the alkylated lactone **248**, stereoselectively. Hydrolysis of **248** provided the corresponding carboxylic acid derivative. To prevent re-lactonization, silylation of the alcohol group with TBDMSCl was performed. The introduction of the P2 amine was achieved via the HATU-mediated coupling. Subsequently, the heterocyclic aromatic group R_3_ (MP or DMAP) was introduced through palladium-assisted Suzuki coupling of boronic acids, producing compounds **250**. Finally, *N*-Boc deprotection under acid conditions was followed by HATU-mediated coupling of *N*-(methoxycarbonyl)-*L*-*tert*-leucine, resulting in the desired HIV-1 PIs **252** with high overall yields and enantioselective purity (*ee* > 95%).

In the last period, studies on the synthesis of key heterocyclic fragments, particularly those oriented towards the pharmaceutical industry, were described. Kappe and his group developed multistep continuous flow reaction method for the synthesis of the biaryl-hydrazine unit of atazanavir [71]. The synthesis involved Pd-catalyzed Suzuki–Miyaura cross-coupling, followed by hydrazone formation and a hydrogenation step. At the end, an additional liquid–liquid extraction step was performed. The method yielded the desired product with an overall yield of 74%, which exceeded the 53% overall batch yield previously described in the literature.

A useful method was developed by Lindhardt and Skrydstrup for the synthesis of active esters by palladium-catalyzed alkoxycarbonylation of (hetero)aromatic bromides (Figure 48) [72]. The protocol was general for a range of oxygen nucleophiles including *N*-hydroxysuccinimide (NHS) **252** and showed high functional group tolerance. The method enabled the synthesis of an important precursor to the HIV protease inhibitor saquinavir, through the formation of an NHS ester followed by acyl substitution.

Quinoline and isoquinoline moieties have been extensively studied as scaffolds for HIV inhibition and have been recently reviewed [73]. In the work of Sarveswari [74], two diverging series of water-soluble, non-peptidic quinoline analogs were designed and synthesized through the sequential attachment of piperazine and various amino acids to the quinoline scaffold using peptide coupling procedures. All synthesized compounds were subjected to in silico screening against HIV protease-1 and the cytotoxicity of all compounds was examined on HCT116 cells. In particular, compound **259** (Figure 49), which bears serine and piperazine in sequence at C2 of quinoline, showed significant HIV protease inhibition properties and demonstrated cytotoxicity IC_50_ value at 22.7 ± 0.59 nM. The synthesis began with the construction of quinoline core via HCl-promoted coupling between ethyl 4-chloroacetoacetate **254** and 2-amino-5-chlorobenzophenone **253** in methanol, yielding 94% of the desired compound **255**. In the second step, Boc-piperazine was introduced by nucleophilic substitution, leading to compound **256**. After removing the Boc protecting group, the piperazine amine was coupled with *L*-serine using HOBt, EDC. Final deprotection of the amino acid in methanolic HCl afforded the target compound **259**.

On the way to extending hydrogen bonding interactions between inhibitor and HIV-1 protease, pyrimidine bases were recognized as suitable as P2 ligands capable of enhancing the activity of the inhibitors. In Wang’s work [75], inhibitor **264** (Figure 50), bearing *N*-2-(2,4-Dioxo-3,4-dihydropyrimidin-1(2H)-yl) acetamide and a 4-methoxylphenylsulfonamide, showed high enzyme inhibitory activity, with an IC_50_ of 2.53 nM in vitro and an inhibition ratio with 68% against wild-type HIV-1 in vivo, with low cytotoxicity. Its antiviral activity was also effective against DRV-resistant HIV-1 variants. The syntheses began with the preparation of substituted 2, 4-dioxopyrimidin-1(2H)-yl acetic acid **262** from uracil, via *N*-alkylation with ethyl bromoacetate (38%), followed by saponification with sodium hydroxide (61%). Coupling with the suitable amine **263** under EDCl/HOBt/DMAP conditions yielded the final product in high yield (98%).

Atazanavir is one of the most prescribed HIV-1 protease inhibitors approved by the FDA. It was the first protease inhibitor approved for once-a-day dosing to treat AIDS, owing to its good oral bioavailability and favorable pharmacokinetic profile. Reddy’s work [76] resulted in a new multistep synthesis for biaryl-hydrazine unit {*tert*-butyl 2-[4-(2-pyridinyl)benzyl]hydrazinecarboxylate} of atazanavir **268** on a large scale (Figure 51). The synthesis began with a palladium catalyzed Suzuki–Miyaura coupling of readily available 2-chloropyridine **264** and (4-cyanophenyl)boronic acid **265**, yielding the biaryl derivative **266**. Next, the cyano group was reduced to the aldehyde **267** using DIBAL-H. Treatment with *tert*-butyl carbazate, followed by in situ reduction with NaBH_4_-furnished hydrazone **268**. The entire process was scaled-up and completed in three steps with an overall yield of 71%.

Several naturally occurring triterpenes and their semisynthetic analogs have demonstrated potent anti-HIV activities. Recently, Zheng [77] synthesized thirteen nitrogen-containing derivatives of 3,11-dioxo-olean-12- en-30-oic acid by introducing various amino acids and nitrogen-containing heterocyclic groups at the 30-carboxyl group, starting from 18*β*-glycyrrhetinic acid (**GA**) (Figure 52). Among them, compound **270** displayed relatively moderate inhibitory activity, with IC_50_ values below 0.24 mM. Molecular docking studies revealed favorable hydrophobic–hydrophobic and hydrogen bonding interactions in the active site of HIV-1PR. These findings underscore the potential of such derivatives as promising candidates for the development of HIV-1 PR inhibitors. The synthesis commenced with the oxidation of the 3-hydroxyl group of **GA** to a carbonyl group (97%), yielding compound **269**. The target compounds **270** were synthesized using an EDCI/HOBt/DMAP/TEA system in the presence of a suitable nucleophile.

Certain thiazolyl and benzothiazolyl guanidines have been reported as exhibiting a wide range of pharmacological and antimicrobial activities. In the field of carbohydrates, glycosyl isothiocyanates are versatile synthetic intermediates for the synthesis of biologically active carbohydrate derivatives. Cao et al. synthesized and evaluated the bioactivity of some new *N*-glucosyl-*N*′-(4-arylthiazol-2-yl) aminoguanidines (Figure 53) [78] The starting materials, 2-amino-4-arylthiazoles of type **271**, were refluxed with substituted glycosyl isothiocyanates in dry benzene. Subsequent desulfurization of the resulting thioureas with HgCl_2_ in DMF, in the presence of hydrazine hydrate and TEA, furnished the target compounds. In particular, compounds **274** and **275** showed moderate activity against HIV-1 protease (IC_50_ between 22 and 107 mg/mL).

While ritonavir is approved as an HIV PI, it is hardly used as such, being more frequently used as a pharmacokinetic enhancer. Unfortunately, its use is often associated with various side effects. Therefore, novel derivatives have been described. Jonckers [79] proposed thiazol-5-ylmethyl (2*S*,3*R*)-4-(2-(ethyl(methyl)amino)-*N*-isobutylbenzo[d]oxazole-6-carboxamido)-3-hydroxy-1-phenylbutan-2-ylcarbamate **281** as a lead candidate for this class (Figure 54). This compound, together with structurally similar analogues, demonstrated excellent ‘boosting’ properties when tested in dogs. These findings made it attractive in the search for novel pharmacokinetic enhancers. The synthesis of the lead compound **279** began with commercially available epoxide **276**, which was reacted with an excess of isobutyl amine, yielding monoprotected *bis*-aminoalcohol **277**. This was then coupled with acid derivative **278** using HATU or BOP as activating agent, yielding intermediate **279**. After mild Boc-deprotection, the intermediate amine was coupled with suitable thiazolyl carbonate **280**, affording target compound **281** in high yield. The in vitro antiviral activity against wild-type HIV-1 (EC_50_ = 71 mM) was evaluated in an acutely infected lymphoblastic cell line (MT4-LTR-EGFP) and CYP 3A4 inhibition (IC_50_ = 0.031 mM) was determined in vitro using a human liver microsome (HLM)-based assay.

Multiple heteroaromatic fragments were introduced into novel inhibitors. In particular, Houpis [80] described the convergent synthesis of clinical candidate **282** (Figure 55), a protease inhibitor specifically designed to allow for long acting-controlled release formulations. Central disconnection generated two synthons **283** and **284** bearing heteroaromatic moieties.

The preparation of epoxide **284** started from protected tyrosine **285** (Figure 56). The phenol functional group was activated as the triflate, using triflic anhydride in the presence of pyridine. The resulting product was then coupled with a suitable pyridyl boronic acid in the presence of commercially available ferrocenyl-based catalyst, Cl_2_Pd (dppf)-DCM, yielding the *bis*-aryl intermediate **286**. The key intermediate, epoxide **284**, could be obtained from either metal-catalyzed or enzymatic process in high yield by exposing **287** to aqueous potassium hydroxide in *tert*-amyl alcohol.

The optically pure amino alcohol (*R*,*R*-**293**) was prepared starting from ketone **289** (Figure 57), which was brominated selectively at the α-aliphatic position using CuBr_2_ in refluxing EtOAc to give *rac*-**290**. An NaBH_4_ reduction followed by a Ritter reaction afforded the racemic *cis*-amino alcohol **292**. The optically active derivative (***R***,***R*-293**) was obtained in good yield via a resolution with *S*-mandelic acid. The optically active amide **283** was prepared using well established methods. Subsequent coupling with epoxide **284** was performed using ZnCl_2_/TMEDA system, followed by the addition of LiHMDS. Deprotection of the BOC group and coupling with the methyl carbamate *tert*-butylglycine furnished the desired inhibitor **282**.

Ghosh [81] described the design, synthesis, and biological evaluation of a series of novel HIV-1 protease inhibitors bearing isophthalamide derivatives as the P2–P3 ligands. In particular, compound **295** showed an enzyme Ki of 0.17 nM and antiviral IC_50_ of 14 nM (Figure 58). The general synthetic strategy involved the coupling of isophthalic acid derivative **296** with the amine of the hydroxyethylamine sulfonamide isosteres **297** to produce HIV-1 protease inhibitors. Various isophthalic acid derivatives can be obtained by coupling readily available isophthalic monoacid **298** with various amines. In particular, amine **299** was prepared from oxazole ester **300**, which was first reduced with excess of DIBAL-H to afford the corresponding alcohol **301**. Subsequently, azidation of the alcohol with diphenylphosphoryl azide in the presence of DBU provided the corresponding azide. Finally, reduction of the azide with triphenyl phosphine in THF yielded amine **299**.

Ghosh and his group described the effect of a single atom change or a scission of a single bond on the activity of a panel of compounds structurally like darunavir. Seven novel PIs were synthesized, and the modifications involved an exchange of sulfur for oxygen, a scission of a single bond in P2′-cyclopropylaminobenzothiazole (or -oxazole), and/or introduction of P1-benzene ring with mono- or *bis*-fluorine atoms (Figure 59) [82]. X-ray structural analyses of the PIs complexed with wild-type protease (PR_WT_) and highly multi-PI-resistant PR_DRV P51_ revealed that the PIs adjusted themselves to the protease with resistance-associated amino acid substitutions. Inhibitors containing a benzothiazole moiety at the P2′ position showed greater anti-HIV-1 activity than those with a benzoxazole moiety. It was claimed that this greater potency is attributed to the capacity of sulfur atoms to form bidirectional σ-hole potentials with the carbonyl oxygen of G48 [83]. On the other hand, substitution of cyclopropyl with isopropyl at the distal part of the inhibitor’s P2′ moiety resulted in a reduction in the antiviral activity. The membrane penetration data confirmed previous findings that the addition of two fluorine atoms greatly boosts the activity of the inhibitors.

More recently, Ghosh et al. [84] described new inhibitors bearing various pyridyl-pyrimidine, aryl thiazole, or alkylthiazole moieties as P2 ligands in darunavir-like hydroxyethylamine sulfonamide isosteres, with the aim of promoting hydrogen bonding interactions with the backbone atoms in the S2 subsite of HIV-1 protease. The different ligands were introduced using suitable parent benzoic acids, whose synthesis was described. The new inhibitors showed sub-nanomolar levels of protease inhibitory activity and low nanomolar levels of antiviral activity. In particular, pyridyl-pyrimidine benzamide derivative **303a** (Figure 60) displayed an enzyme inhibitory K*i* of 28 pM and antiviral activity of 154 nM, whereas compound **303b** displayed potent antiviral activity with an IC_50_ value of 66 nM. Among the thiazole-derived inhibitors, compound **308**, with a pyridyl thiazole as the P2 ligand, showed the best result, exhibiting an HIV-1 protease inhibitory *K*i of 8.7 nM and antiviral activity of 580 nM.

For the synthesis of **303**, commercially available 4-methyl-3-[[4-(3- pyridinyl)-2-pyrimidinyl]amino]benzoic acid **302** was reacted with the previously reported hydroxyethylene isosteres **297** in the presence of HATU and Et_3_N in DMF. The synthesis of **308** started from 2-methyl 5-aminomethylbenzoate **304**, which was reacted with potassium thiocyanate and acetyl chloride in acetone to yield the thiourea derivative **305**. Exposure of **305** to potassium carbonate in methanol, followed by the addition of *α*-bromo ketone **306**, furnished the pyridinylthiazole derivative **307**. Saponification of methyl ester with aqueous LiOH in THF afforded the corresponding carboxylic acid. Final coupling with the hydroxyethylsulfonamide isosteric amine **297a** produced inhibitor **308** in good yields.

In 2010, we synthesized a series of new thienyl analogues of nelfinavir and saquinavir with different substitution patterns, derived from suitable enantiopure diols [85]. Their inhibitory activity against wild-type recombinant HIV-1 protease was evaluated. In general, thienyl groups spaced from the core by a methylene group yielded products with IC_50_ values in the nanomolar range, regardless of the heterocycle’s type or substitution pattern. Notably, compounds **309** and **310** (Figure 61) were the most active, and their activity was substantially maintained or even increased against two common mutants under drug pressure, such as V32I and V82A. The synthesis began with dihydroxybutylesters **311**, which were reduced by BH_3_SMe_2_, producing triols **312** in nearly quantitative yield. Freshly prepared 3,3-dimethoxypentane was used for selective protection of the hydroxyl groups on carbons 1 and 2. The free hydroxyl was then transformed into the mesylate **313** and subsequently substituted with an azido group. Final hydrolysis yielded azidodiols **314** with modest to good overall yield.

The perhydroisoquinolinic fragment was linked via selective activation of the hydroxyl group as a mesitylene sulfonyl derivative **316**, followed by reaction with the commercially available substituted perhydroisoquinoline (Figure 62). Finally, the azidoalcohols were transformed into amino alcohols **318a** and **318b** by Pd-catalyzed hydrogenation in excellent yield.

For the synthesis of the nelfinavir thienyl analog, amino alcohol **318a** was reacted with 3-acetoxy-2-methylbenzoic acid. Final deacetylation of the phenolic group yielded the target compounds **309** in good yield (Figure 63). The characteristic dipeptide unit of saquinavir derivatives **319** was prepared by coupling asparagine tert-butyl ester with quinolinic acid. Hydrolysis with TFA afforded acid **319**, which was subsequently coupled with amino alcohol **318b**, resulting in the desired saquinavir derivatives **310**.

Investigating the synthesis and biological evaluation of new structurally simplified non-peptidic heteroaromatic molecules as PIs, we described the synthesis and biological evaluation of a new series of potential HIV-1 protease inhibitors of types **330**–**333**, incorporating different benzofused heterocycles (Figure 64) [86]. The variation in heteroatoms in such molecules affected biological activities and a benzothiophene containing inhibitor **333** exhibited high potency against wild-type HIV-1 protease with an IC_50_ of 60 nM, thanks to the lower desolvation penalty paid by this hydrophobic moiety. The synthesis began with commercially available (*S*)-glycidol (98% *ee*), which was reacted with 3-nitrobenzenesulfonyl chloride (NsCl) and triethylamine (Et_3_N) at −10 °C to afford **320** in 80% yield. Subsequently, it underwent regioselective displacement of the nosylate with the appropriate 5-hydroxyheteroarene and K_2_CO_3_ to yield the corresponding epoxides **321**, **324**, and **325** in 70%, 78%, and 82% yield, respectively. The indole derivative **321** was alkylated with methyl iodide and benzyl chloride to produce compounds **322** and **323** in 90% and 86% yield. The opening of the oxiranyl ring with *i*-BuNH_2_ in *i*-PrOH provided aminoalcohols **326**–**329** in quantitative yield. These compounds were then reacted with 3,4-dimethoxybenzenesulfonyl chloride and Et_3_N in dry DCM to afford target compounds **330**–**333** in high yield.

With the aim of facilitating access to new HIV-1 protease inhibitors bearing heteroaryl moieties as P1-ligands, we speculated on a convenient synthetic route to introduce diversity into the common hydroxyethylamino core present in several approved PIs [87] In a straightforward retrosynthetic approach, variously functionalized aromatic groups could be incorporated via Suzuki coupling between an activated C(sp3) bromide (allylic electrophile) and an array of arylboronic acids, thereby furnishing methyl 4-arylcrotonates. An effective ligand-free Suzuki coupling protocol was described for coupling methyl (*E*)-4-bromobut-2-enoate with several arylboronic acids. Given the strong interest in methodologies that produce polyarylated frameworks, we subsequently reported a nickel-catalyzed double phenylation of methyl 4-bromocrotonate, which furnished suitable doubly phenylated building blocks [88].

Different aspects regarding the preparation of peptidomimetic and pseudopeptidic structures containing heterocycles were also reviewed in 2012 by us, with particular focus on novel tricyclic structures as potential drugs [89]. Following the concept of targeting the protein backbone, we systematically investigated various substitution patterns on the common stereo-defined isopropanolamine core. In 2014, we described the synthesis of new, structurally simple indolic non-peptidic HIV-protease inhibitors from (S)-glycidol using regioselective methods [90], varying the type and/or the position of the functional group on the indole and the nature of the nitrogen containing group (sulfonamides or perhydroisoquinoline). The systematic study of in vitro inhibition activity of these compounds confirmed the general beneficial effect of the 5-indolyl substituents in the presence of arylsulfonamide moieties, which showed activities in the micromolar range. Oxyindoles and carbamoyl indoles showed general good activity, whereas simple aminoindoles were much less active (Figure 65). For the synthesis of oxyindoles, epoxide **322** was opened with perhydroisoquinoline, yielding inhibitor **334**, or alternatively with *i*-BuNH_2_, giving intermediate **338**. This last amino alcohol was reacted with 3,4-diOMe-phenylsulfonyl chloride, affording compound **335** in excellent yield.

The preparation of the corresponding carbamoyl derivatives **336** and **337** was straightforward. 5-Aminoindole **339** was first reacted with *p*-nitrophenylchlorocarbonate to afford the activated carbamate **340** (Figure 66). Glycidol was then introduced via a substitution reaction, yielding the oxiranyl carbamate **341** in good yield. The introduction of *i*-BuNH_2_, followed by sulfonylation with a suitable ArSO_2_Cl, furnished the final compounds **336** and **337**.

Different spacers were studied, connecting heteroaryl moieties to the hydroxyethylamine core. Thus, in 2017, new heteroaryl HIV protease inhibitors, bearing a carboxy–amide spacer, were synthesized in our lab in a few steps and with high yield, starting from commercially available homochiral epoxides [91]. Onto a given hydroxyethyl-amino-isopropanoyl-sulfonamide core, we introduced different heteroarenes and modified the central core, with the presence of either H or a benzyl group (structure **A** in Figure 67). For the synthesis of the simple, unsubstituted isopropanolamine core (R = H), we took advantage of an established route, starting from the commercially available bidentate electrophile (S)-glycidol. The epoxide was opened by *i*-PrNH_2_; the aryl sulfonyl moiety was introduced, and the primary OH group was substituted by NH_2_, furnishing the key aminoalcohol **343** in good overall yield. Coupling with suitable 5-heteroaryl acids afforded the corresponding amides **344**.

The synthesis of benzyl derivatives **347** was even shorter (Figure 68) and started from the commercially available homochiral *N*-Boc-protected amino-epoxide **345**. After opening with *i*BuNH_2_ and the subsequent introduction of the arylsulfonyl moiety, the *N*-Boc group was displaced by treatment with trifluoroacetic acid in dichloromethane. The resulting ammonium trifluoroacetate was treated with NEt_3_, affording amine **346**, which was then reacted with suitable 5-heteroarylcarboxylic acids previously activated with *N*,*N*′-carbonyldiimidazole. Thus, the final products **347** were obtained in four steps and excellent overall yield.

In general, the presence of a carboxyamide moiety showed a positive effect on in vitro inhibition activity against recombinant protease. The IC_50_ values ranged between 1 and 15 nM. In particular, benzofuryl derivatives **344a** and **347a** displayed some of the best IC_50_ values among such structurally simple inhibitors. Docking analysis supported the experimental results regarding activity, demonstrating that these benzofuryl derivatives exhibited a favorable number of interactions within the active site. The inhibitory activity of these molecules was also evaluated in HEK293 cells.

The study of new heteroaryl HIV protease inhibitors was extended to those bearing a carbamoyl spacer [92]. In particular, we focused on new derivatives with the general structure **B** (Figure 69), in which the heterocycle is spaced from the core by a carbamoyl function, like the arrangement in darunavir and TMC-126. Their synthesis was straightforward from commercially available homochiral epoxides. Different substitution patterns were introduced onto a given isopropanoyl-sulfonamide *core*, which could have either H or benzyl group.

Both the carbamoyl moiety and benzyl group displayed a general beneficial effect on the in vitro inhibition activity against recombinant protease. The IC_50_ values ranged from 11 to 0.6 nM. In particular, benzofuryl and indolyl derivatives **351a** and **351c** showed some of the best IC_50_ values (IC_50_ 0.6 nM for both). Regarding the amide inhibitors, their activity was also confirmed in HEK293 mammalian cells and was maintained against protease mutants. Furthermore, the metabolic stability of all compounds was studied and found to be comparable to that of commercially available inhibitors. More recently, following the concept of repositioning HIV protease inhibitors as cancer therapeutics, compound **349b** was also evaluated for its ability to induce cytotoxicity in hepatocellular carcinoma cell lines [93].

Considering the structure of HIV protease as a C2-symmetric homodimer in its active form, we recently reported on the synthesis, enzyme inhibition, and structure–activity relationship of a new class of HIV-1 protease inhibitors containing a pseudo-symmetric hydroxyethylamine core and heteroarylcarboxyamide moieties [94]. To obtain a pseudo-symmetric hydroxyethylamine core, a benzyl group was placed on the sulfonamide nitrogen, resulting in the general structure **C** (Figure 70). A straightforward synthetic pathway yielded nine compounds in a few steps with high yields. Potent inhibitory activity, with nanomolar IC50 values measured with a standard fluorimetric test, was achieved. In particular, compounds **353a**–**c** (whose synthesis is described in Figure 70), which contain the indole ring in P1, exhibited HIV-1 protease inhibitory activity that was more potent than darunavir in the same assay.

Recently, considering all our data on heteroaryl non-peptidic inhibitors, specifically their high HIV protease inhibitory activity and easy of synthetic approach, we reported novel simple heteroaryl carboxamides (Figure 71), bearing *p*-NO_2_ electron withdrawing group or *p*-OMe electron releasing group on arylsulfonamide. We compared their in vitro activity in HEK293 cells with that of our previously described compounds [95].

Benzofuryl-, benzothienyl-, and indolyl rings were introduced via efficient synthetic procedures. All compounds showed inhibitory activity comparable to the commercial drug darunavir, with particularly potent examples such as **356a**, **356b**, and **355b** (IC_50_ < 0.6 nM). These compounds were effective against both wild-type HIV-1 protease and mutants containing V32I or V82A mutations. In silico evaluation of their absorption, distribution, metabolism, and excretion (ADME) properties was also conducted, comparing the results with the predicted properties of darunavir. As a result, 12 out 27 compounds (including **355b**) performed better than or equal to darunavir across all ADME prediction models, demonstrating the potential of these compounds for further drug development.

## 3. Conclusions

In conclusion, the introduction of heterocyclic moieties into HIV protease inhibitor scaffolds remains a powerful tool to enhance their activity and overcome enzyme mutations. Both non-aromatic heterocycles, owing to their rigid structures, and heteroaromatics, due to their planar structures combined with modulation of hydrogen bonding interactions, can expand the pool of inhibitors by following the concept of targeting the protein backbone.

Although much has been described regarding the interaction between heterocyclic fragments and the enzyme, the field remains open to further investigations, particularly concerning heterocycles with multiple heteroatoms (aromatic and non-aromatic) and simplified structures that reduce synthetic costs.

Table 1 summarizes the most significant new HIV-protease inhibitors appearing in the literature during the last 15 years.

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
