# Peer review of "Recent Advances in Heterocyclic HIV Protease Inhibitors"

_ijms, 2025, doi:10.3390/ijms26189023_

Round 1
Reviewer 1 Report
Comments and Suggestions for Authors
In the manuscript, the authors tend to introduce the synthesis and biological activity of the heterocyclic HIV protease inhibitors (ynthesis is mainly focused). In fact, the authors provided many synthetic routes. However, synthetic routes. However, after reading the whole review manuscript, it reads more like an accumulation of knowledge instead of a review article with good summary and insight. Therefore, the review manuscript is not recommended to be accepted by the International Journal of Molecular Sciences. Here are some concerns for the authors to improve the quality which may help the authors for the possible next submission to other journals:
1) In the Abstract section, this section reads not finished. I recommend the authors add one period and several sentences to introduce the work of the review manuscript.
2) For the Introduction section, the full name of FDA should be provided when it first appears in the manuscript.
3) For the last paragraph of the Introduction section, the authors should add one sentence to summary the meaning of the review work.
4) I strongly recommend the authors provide one Table including the classification of the introduced chemicals including the category, present biological activities, references, chemical structures for better reading and understanding.
5) The authors should revise each subtitle of section 2 according to the classification.
6) At least one paragraph should be added before the last paragraph of the manuscript to provide perspectives of such field.
Overall, even though the authors provided enough schemes regarding the synthesis of heterocyclic HIV protease inhibitors, the review manuscript can not reach the criteria of the International Journal of Molecular Sciences as it only contains the mere accumulation of knowledge.
Author Response
Comment 1) In the Abstract section, this section reads not finished. I recommend the authors add one period and several sentences to introduce the work of the review manuscript.
Answer 1) The abstract was re-written following the referee’s suggestions.
Comment 2) For the Introduction section, the full name of FDA should be provided when it first appears in the manuscript.
Answer 2) The full name of the FDA has been provided at its first appearance in the Introduction section.
Comment 3) For the last paragraph of the Introduction section, the authors should add one sentence to summary the meaning of the review work.
Answer 3) The introduction has been modified and a last sentence has been added.
Comment 4) I strongly recommend the authors provide one Table including the classification of the introduced chemicals including the category, present biological activities, references, chemical structures for better reading and understanding.
Answer 4) A final table with the most significant compounds described has been added.
Comment 5) The authors should revise each subtitle of section 2 according to the classification.
Answer 5) Subtitles have been modified in a more general form (Non-aromatic O-heterocycles, Non-aromatic N-heterocycles, Non-aromatic heterocycles with multiple heteroatoms, Heteroaromatics).
Comment 6) At least one paragraph should be added before the last paragraph of the manuscript to provide perspectives of such field.
Answer 6) A final paragraph of general perspective has been added.
Reviewer 2 Report
Comments and Suggestions for Authors
This review article details advances made in the field of HIV protease inhibitors covering the past 15 years, focusing in detail on synthetic pathways and biological activity of compounds containing heterocyclic ring systems. In addition, the authors provide in many cases the rationale behind compound design where the heterocyclic system plays a key role in binding to the active site.
The article is well-organized and categorized based on type of heterocyclic ring system used and quite detailed pertaining to the preparative routes of the compound. Comparisons of biological activity versus commercial standards for most of the classes of compounds are also described. The authors are to be commended for the thoroughness of reference citations.
There are minor editing changes required throughout the text that are required and detailed as line number citations below; however, this does not detract from the importance of this review. As such, I recommend publishing with minor changes.
Line 11: first sentence change to "The first cases of AIDS, an important disease caused by the virus HIV-1, was reported in 1980.'
Line 16: insert the word "as" after the word "appeared"
Line 22: typo - change end to "acquired immunodeficiency syndrome"
Line 23: typo - change to "disruption" and replace "Linfocites T" to "CD4+T cells"
Line 26: insert dash ("non-dividing")
Line 37: "Through the incorporation of the hydroxyethylene core,....."
Line 47: insert the words "illustrated with" between "as" and "FDA"
Line 55: change "The" to "This"
Line 56: lower case p in "protease"; change "or" to "and"
Line 68: remove period after the word "by"
Line 75 and thereafter throughout manuscript: regarding incorporation of the word "Scheme XX" in the paper - in most cases the S is not capitalized; in some cases it is capitalized. For continuity, I suggest the S should be capitalized
Line 92: insert comma after the word "recently"
Line 93: K in "ß-Ketolactone" should be capitalized
Line 151: change wording to "Shortly thereafter"
Line 167: capitalize G in (S)-Glyceraldehyde
Line 170: after "Scheme 10", eliminate the words "It was" thereby bridging the 2 sentences
Line 188: change "a,b" to "α,ß...."
Line 203: add the word "materials" after the word "pool"
Line 208: change to "base-mediated"
Line 235: change wording to "with H2O to afford....."
Line 236: change wording to "....as the TBS ether,....."
Line 270: change to "difluorophenylmethyl"
Line 303: delete comma after "recently" and "material" and add the word "the" before "taking" (line 304)
Line 316: delete comma after"reduction"
Line 342: insert comma after the word "methoxide"
Lines 423-424: "Ring closing metathesis..."
Line 439: typo - should be "opened"
Line 456: Should be placed under Scheme 28 on Line 455
Line 461: insert "the" after the word "heterocycles,......."
Line 482: delete "and they" and exchange for "which"
Line 483: in silico
Line 495: should be "N-Tosyl"
Line 513: Scheme 32 should be place under Scheme at Line 512
Line 521: remove extra period
Line 524: should read: ".....enamine 164 which was reacted with benzaldehyde to obtain the monomer 165."
Line 547: start sentence with the word "The"
Line 554: drop the word "the"
Line 562: change to "Piperidines and Pyrrolidines"
Line 581: change to "(Figure 1)"
Line 589: not a Scheme; shoud be captioned "Fig. 1"
Line 593: change to "The five membered indantoine scaffold was already recognized as a key...."
Line 602: via
Line 608: insert "a" after "as"
Line 624: insert caption at Line 623
Line 640: remove period and words "This last" and add "which"
Line 689: insert the word "a" before "suitable"
Line 722: in silico
Line 749: "derivatized"
Line 758: "discovery"
Line 786: change to ".....fragments, those more...."
Line 787: insert comma after "industry" change to "Kappe and his group...." (delete "Oliver")
Line 792: delete "competed to" and add the word "exceeded"
Lines 844-845: "The work of Reddy [68] resulted in...."
Line 860: insert comma after "them"
Line 866" change to "The target compounds 270...."
Line 876: "......derivatives. Cao et. al......"
Line 888: ".....frequently used as a ......"
Line 895: delete comma after "276"
Line 898: delete the word "reaction"
Line 899: delete the word "to"
Line 905: add to title "Compound 281"
Line 914: change to "approach"
Line 917: insert the word "the" after "as"
Line 918: "with a suitable...."
Line 930: via
Line 944: delete "would" and add "can"
Line 955: delete "In a recent work...."
Line 961: lower case p for protease
Line 965: ".....claimed that the greater potency is attributed to...."
Line 974: "more recently, Gosh, et. al. [76]...."
Line 976: change "promote" to "promoting"
Line 992: delete "already" and replace with "previously"
Line 1009: mention "dihydroxybutyl esters 311" but structure illustrated is not numbered in the Scheme 62 and is an α, β unsturated ester - please clarify
Line 1034: structure 310 is missing an oxygen atom
Line 1036: delete the word "on"
Line 1042: delete the words "to be"
Line 1071: typo (systematically)
Line 1089: "5-Aminoindole....."
Line 1101: change to "Onto"
Line 1133: ".....bearing a carbonyl...."
Line 1145: "amide inhibitors"
Line 1166: "non-peptidic"; also, delete comma after the word "inhibitors"
Line 1178: in silico
Line 1184: Insert as a separate line item the word "Conclusion"
Reference Section: All references including titles should be described in sentences with appropriate capitalization. The authors have blended styles such that some references (number 7, Line 1209ff) are correct, while others (number 22) are incorrect based on the ACS Style Guide. Please normalize all references.
Comments on the Quality of English LanguageFor the most part, the English is adequate.
Author Response
There are minor editing changes required throughout the text that are required and detailed as line number citations below; however, this does not detract from the importance of this review. As such, I recommend publishing with minor changes.
Line 11: first sentence change to "The first cases of AIDS, an important disease caused by the virus HIV-1, was reported in 1980.'
Line 16: insert the word "as" after the word "appeared"
Line 22: typo - change end to "acquired immunodeficiency syndrome"
Line 23: typo - change to "disruption" and replace "Linfocites T" to "CD4+T cells"
Line 26: insert dash ("non-dividing")
Line 37: "Through the incorporation of the hydroxyethylene core,....."
Line 47: insert the words "illustrated with" between "as" and "FDA"
Line 55: change "The" to "This"
Line 56: lower case p in "protease"; change "or" to "and"
Line 68: remove period after the word "by"
Line 75 and thereafter throughout manuscript: regarding incorporation of the word "Scheme XX" in the paper - in most cases the S is not capitalized; in some cases it is capitalized. For continuity, I suggest the S should be capitalized
Line 92: insert comma after the word "recently"
Line 93: K in "ß-Ketolactone" should be capitalized
Line 151: change wording to "Shortly thereafter"
Line 167: capitalize G in (S)-Glyceraldehyde
Line 170: after "Scheme 10", eliminate the words "It was" thereby bridging the 2 sentences
Line 188: change "a,b" to "α,ß...."
Line 203: add the word "materials" after the word "pool"
Line 208: change to "base-mediated"
Line 235: change wording to "with H2O to afford....."
Line 236: change wording to "....as the TBS ether,....."
Line 270: change to "difluorophenylmethyl"
Answer 1) All the changes suggested by the reviewer have been implemented.
Line 303: delete comma after "recently" and "material" and add the word "the" before "taking" (line 304)
Answer 2) The sentence “An alternative route was proposed more recently starting from inexpensive material taking advantage of the highly enantioselective enzymatic desymmetrization” was changed to “A more recent alternative route has been proposed, starting from inexpensive material and taking advantage of highly enantioselective enzymatic desymmetrization”
Line 316: delete comma after"reduction"
Line 342: insert comma after the word "methoxide"
Lines 423-424: "Ring closing metathesis..."
Line 439: typo - should be "opened"
Line 456: Should be placed under Scheme 28 on Line 455
Line 461: insert "the" after the word "heterocycles,......."
Line 482: delete "and they" and exchange for "which"
Line 483: in silico
Line 495: should be "N-Tosyl"
Line 513: Scheme 32 should be place under Scheme at Line 512
Line 521: remove extra period
Line 524: should read: ".....enamine 164 which was reacted with benzaldehyde to obtain the monomer 165."
Line 547: start sentence with the word "The"
Line 554: drop the word "the"
Line 562: change to "Piperidines and Pyrrolidines"
Line 581: change to "(Figure 1)"
Line 589: not a Scheme; shoud be captioned "Fig. 1"
Line 593: change to "The five membered indantoine scaffold was already recognized as a key...."
Line 602: via
Line 608: insert "a" after "as"
Line 624: insert caption at Line 623
Line 640: remove period and words "This last" and add "which"
Line 689: insert the word "a" before "suitable"
Line 722: in silico
Line 749: "derivatized"
Line 758: "discovery"
Line 786: change to ".....fragments, those more...."
Line 787: insert comma after "industry" change to "Kappe and his group...." (delete "Oliver")
Line 792: delete "competed to" and add the word "exceeded"
Lines 844-845: "The work of Reddy [68] resulted in...."
Line 860: insert comma after "them"
Line 866" change to "The target compounds 270...."
Line 876: "......derivatives. Cao et. al......"
Line 888: ".....frequently used as a ......"
Line 895: delete comma after "276"
Line 898: delete the word "reaction"
Line 899: delete the word "to"
Line 905: add to title "Compound 281"
Line 914: change to "approach"
Line 917: insert the word "the" after "as"
Line 918: "with a suitable...."
Line 930: via
Line 944: delete "would" and add "can"
Line 955: delete "In a recent work...."
Line 961: lower case p for protease
Line 965: ".....claimed that the greater potency is attributed to...."
Line 974: "more recently, Gosh, et. al. [76]...."
Line 976: change "promote" to "promoting"
Line 992: delete "already" and replace with "previously"
All the changes suggested by the reviewer have been implemented.
Line 1009: mention "dihydroxybutyl esters 311" but structure illustrated is not numbered in the Scheme 61 and is an α, β unsturated ester - please clarify.
Answer 2) The scheme 61 has been corrected
Line 1034: structure 310 is missing an oxygen atom
Line 1036: delete the word "on"
Line 1042: delete the words "to be"
Line 1071: typo (systematically)
Line 1089: "5-Aminoindole....."
Line 1101: change to "Onto"
Line 1133: ".....bearing a carbonyl...."
Line 1145: "amide inhibitors"
Line 1166: "non-peptidic"; also, delete comma after the word "inhibitors"
Line 1178: in silico
Line 1184: Insert as a separate line item the word "Conclusion"
Reference Section: All references including titles should be described in sentences with appropriate capitalization. The authors have blended styles such that some references (number 7, Line 1209ff) are correct, while others (number 22) are incorrect based on the ACS Style Guide. Please normalize all references.
All the changes suggested by the reviewer have been implemented.
Reviewer 3 Report
Comments and Suggestions for Authors
The structure of the entire manuscript needs to be changed as it contains numerous inconsistencies and logical errors.
During the modification of the manuscript, I would first like to recommend to the authors to avoid using scientific slang, which is critical to the accuracy of the main points of the paper. Here are the exact comments:
- Darunavir and THF derivatives; no way. I strongly recommend not to call hexahydrofuro[2,3-b]furan fragment THF. It is not correct at all, as THF is tetrahydrofuran. A renaming is needed.
- A comprehensive list of abbreviations is absolutely necessary. The text contains a number of uncommon abbreviations, such as KRED300-H2, GDH glucose, KPi etc., which are not widely used. To understand the text, they have to be interpreted.
- Radicals and functional groups should be identified uniformly. If methyl is labeled Me, then it should be labeled that way everywhere, not CH3. The same applies to other radicals. The work is very lacking in the neatness of the design of reaction schemes, in general.
- The same may be applied to the indication of the reagents, in general. In one case lithium aluminium hydride is LAH, but in another ot is LiAlH4 (by the way, Li[AlH4] is more correct). Toluene is toluene in some schemes, but in the others it is PhMe. Just the same with para-toluenesulfonic acid, which is marked as PTSA many times, while it is commonly indicated as TsOH. There are numerous examples of such diversity in the whole text.
- Each text fragment should end with a conclusion and generalization. In the current version, the sections look like a simple enumeration of synthetic circuits, devoid of any conclusions.
- Cyclic ureas: no go. Urea is an acyclic compound. Please use the correct collective names for corresponding heterocycles, such as 1,3-diazacycloalkan-2-ones. By the way 2,6-diphenylpiperidin-4-ones are foreign to this section: thay have to be removed or presented as a separate section.
- Enamino oxindoles: no way. Substituted indolin-2-ones sounds more correct.
- Methyl imine could be replaced by a phenyl imine, furnishing different compounds: what does it mean? There are no such organic copmounds. The sense of this phrase is difficult to understand.
- Piperidines: this part contains the description of cyclooligopeptides, devoid of piperidine ring in their structure. What does it mean? By the way, the info about 2,6-diphenylpiperidin-4-one may look better in this part of the text, just because of the presence of a piperidine ring in the title compunds.
- Idantoins: may be hydantoins? In this part of the text the synthesis of the compund 196 (and 206, as well) is presented. In fact, it is not a hydantoin derivative, but a derivative of 2-amino-1,5-dihydro-4H-imidazol-4-one. Guess, it is necessary to use another name for this part of the text.
- Polyheterocycles: too general. Moreover, BTG(O)-A is not understanable at all.
- Heteroaromatics: no go. There are a number of compounds in this part of the text, which do not contain the heteroaromatic rings. For example, 241 and 242.
- Quinolines, Thiazoles: they are not heteroaromatics? Whay are they presented in a separate part of the work? And then, a plethora of different heterocyles (benzothophene, indole and so on) appears, which are far away from the above mentioned quinoline and thiazole.
Dear authors, in my point of view, tout est à refaire (the manuscript has to be overwritten or, at least, reconstructed). The starting material is sufficient, but its presentation and structure of the manuscript are not acceptable.
Author Response
The structure of the entire manuscript needs to be changed as it contains numerous inconsistencies and logical errors.
During the modification of the manuscript, I would first like to recommend to the authors to avoid using scientific slang, which is critical to the accuracy of the main points of the paper. Here are the exact comments:
Comment 1) Darunavir and THF derivatives; no way. I strongly recommend not to call hexahydrofuro[2,3-b]furan fragment THF. It is not correct at all, as THF is tetrahydrofuran. A renaming is needed
Answer 1) To avoid confusion, the term “THF” has been replaced by “bis-THF”, as this is the commonly used name in the literature for the hexahydrofuro[2,3-b]furan fragment (see: https://doi.org/10.1021/ol703061u).
Comment 2) A comprehensive list of abbreviations is absolutely necessary. The text contains a number of uncommon abbreviations, such as KRED300-H2, GDH glucose, KPi etc., which are not widely used. To understand the text, they have to be interpreted.
Answer 2) To improve clarity, the uncommon abbreviations mentioned (such as KRED300-H2, GDH glucose, KPi, etc.) have been spelled out in the text.
Comment 3) Radicals and functional groups should be identified uniformly. If methyl is labeled Me, then it should be labeled that way everywhere, not CH3. The same applies to other radicals. The work is very lacking in the neatness of the design of reaction schemes, in general.
Answer 3) The radicals have been standardized across all schemes.
Comment 4) The same may be applied to the indication of the reagents, in general. In one case lithium aluminium hydride is LAH, but in another ot is LiAlH4 (by the way, Li[AlH4] is more correct). Toluene is toluene in some schemes, but in the others it is PhMe. Just the same with para-toluenesulfonic acid, which is marked as PTSA many times, while it is commonly indicated as TsOH. There are numerous examples of such diversity in the whole text.
Answer 4) The reagents have been standardized throughout the text and schemes, and the examples mentioned (e.g., LiAlHâ‚„, toluene, TsOH) have been uniformly indicated
Comment 5) Each text fragment should end with a conclusion and generalization. In the current version, the sections look like a simple enumeration of synthetic circuits, devoid of any conclusions.
Answer 5) We consider a conclusion and generalization at the end of any single text fragment not necessary and out of the scope of a review.
Comment 6) Cyclic ureas: no go. Urea is an acyclic compound. Please use the correct collective names for corresponding heterocycles, such as 1,3-diazacycloalkan-2-ones. By the way 2,6-diphenylpiperidin-4-ones are foreign to this section: thay have to be removed or presented as a separate section.
Answer 6) Following the reviewer’s comment, the term cyclic urea has been replaced with the term 1,3-diazacycloalkan-2-one. The section has been re-named as “Non-aromatic N-heterocycles”.
Comment 7) Enamino oxindoles: no way. Substituted indolin-2-ones sounds more correct.
Answer 7) Following the reviewer’s comment, the term enamino oxindoles has been replaced with the term indolin-2-ones.
Comment 8) Methyl imine could be replaced by a phenyl imine, furnishing different compounds: what does it mean? There are no such organic copmounds. The sense of this phrase is difficult to understand.
Answer 8) The meaning of the sentence is that it is possible to start from a different substrate bearing a phenyl group instead of the methyl group (a benzophenone instead of an acetophenone) leading to a series of analogous compounds, as reported in the literature (doi: 10.1016/j.bmc.2015.09.002).
Comment 9) Piperidines: this part contains the description of cyclooligopeptides, devoid of piperidine ring in their structure. What does it mean? By the way, the info about 2,6-diphenylpiperidin-4-one may look better in this part of the text, just because of the presence of a piperidine ring in the title compunds.
Answer 9) The section has been re-named as “Non-aromatic N-heterocycles”.
Comment 10) Idantoins: may be hydantoins? In this part of the text the synthesis of the compund 196 (and 206, as well) is presented. In fact, it is not a hydantoin derivative, but a derivative of 2-amino-1,5-dihydro-4H-imidazol-4-one. Guess, it is necessary to use another name for this part of the text.
Answer 10) The refuses Idantoin have been corrected. For 196 and 206 we kept the nomenclature chosen by the authors (see ref. [51] and [52a].
Comment 11) Polyheterocycles: too general. Moreover, BTG(O)-A is not understanable at all.
Answer 11) The section has been re-named as “Non-aromatic heterocycles with multiple heteroatoms”. The terms “(bicyclices derived from tartaric acid and glycine)” for BTG(O)-A and (bicyclices derived from tartaric acid and phenylalanine) foir BTF(O)-B were added for a better comprehension.
Comment 12) Heteroaromatics: no go. There are a number of compounds in this part of the text, which do not contain the heteroaromatic rings. For example, 241 and 242.
Answer 12) Compound 242 is pyrimidinylthio derivative, thus heteroaromatic, while 241 is cited as aromatic reference compound and is not heteroaromatic.
Comment 13) Quinolines, Thiazoles: they are not heteroaromatics? Whay are they presented in a separate part of the work? And then, a plethora of different heterocyles (benzothophene, indole and so on) appears, which are far away from the above mentioned quinoline and thiazole.
Answer 13) Sections “Quinolines”, “Thiazoles” and so on were suppressed.
Round 2
Reviewer 1 Report
Comments and Suggestions for Authors
All the concerns have been revised all explained by the authros. And the present verision of the manuscirpt has been improved by this round of revision especially for the added table which may help the manuscript read more clearly. The present version of the manuscript is recommended to be accepted by the International Journal of Molecular Sciences. Congratulations to the auhtors!